# Cortical Bone Trajectory Instrumentation with Vertebroplasty for Osteoporotic Thoracolumbar Compression Fracture

**DOI:** 10.3390/medicina56020082

**Published:** 2020-02-17

**Authors:** Wei-Lin Hsu, Yu-Hsiang Lin, Hao-Yu Chuang, Han-Chung Lee, Der-Cherng Chen, Yen-Tse Chu, Der-Yang Cho, Chao-Hsuan Chen

**Affiliations:** 1Department of Neurosurgery, China Medical University Hospital, Taichung 404, Taiwan; altty15680@gmail.com (W.-L.H.); magiclin117@gmail.com (Y.-H.L.); greeberg1975@gmail.com (H.-Y.C.); braintumorgbm@gmail.com (H.-C.L.); vincen1966@gmail.com (D.-C.C.); d10780@gmail.com (Y.-T.C.); d5057@mail.cmuh.org.tw (D.-Y.C.); 2Department of Neurosurgery, Tainan Municipal An-Nan Hospital-China Medical University, Tainan 709, Taiwan

**Keywords:** cortical bone trajectory, osteoporosis, vertebral compression fracture, vertebroplasty

## Abstract

Background: Osteoporotic spinal fractures commonly occur in elderly patients with low bone mineral density. In these cases, percutaneous vertebroplasty or percutaneous kyphoplasty can provide significant pain relief and improve mobility. However, studies have reported both the recurrence of vertebral compression fractures at the index level after vertebroplasty and the development of new vertebral fractures at the adjacent level that occur without any additional trauma. Pedicle screw fixation combined with percutaneous vertebroplasty has been proposed as an effective procedure for addressing osteoporotic thoracolumbar fractures. However, in osteoporotic populations, pedicle screws can loosen, pullout, or migrate. Currently, the efficacy of cortical bone trajectory screw fixation for osteoporotic fractures remains unclear. Thus, we assessed the effects of using cortical bone trajectory instrumentation with vertebroplasty on patient outcomes. Method: We retrospectively reviewed data from 12 consecutively sampled osteoporotic thoracolumbar fracture patients who underwent cortical bone trajectory instrumentation with vertebroplasty. Patients were enrolled beginning in October 2015 and were followed for >24 months. Result: The average age was 74 years, and the average dual-energy x-ray absorptiometry T-score was −3.6. The average visual analog scale pain scores improved from 8 to 2.5 after surgery. The average blood loss was 36.25 mL. All patients regained ambulation and experienced reduced pain post-surgery. No recurrent fractures or instrument failures were recorded during follow-up. Conclusions: Our findings suggest that cortical bone trajectory instrumentation combined with percutaneous vertebroplasty may be a good option for treating osteoporotic thoracolumbar fractures, as it can prevent recurrent vertebral fractures or related kyphosis in sagittal alignment.

## 1. Introduction

Owing to the increasing size of the aging population worldwide, osteoporosis and its associated fractures have become an important health issue. Patients with poor bone mineral density have a high risk of developing osteoporotic vertebral compression fractures (VCFs) from relatively small axial loadings, such as after a trivial fall. During the early vertebral healing period, elderly patients may experience severe motion pain and disability and be bed-ridden, which in turn can lead to severe complications. Moreover, patients’ decreased mobility and restricted activity may further lead to progressive bone density loss, predisposing them to additional kyphosis [1].

Percutaneous vertebroplasty (PVP) or percutaneous kyphoplasty (PKP) can stabilize the fractured vertebral body through the minimally invasive injection of bone cement. Many studies have reported that PVP and PKP significantly relieve back pain via a thermal reaction produced by the cement that blocks nerve endings in the fractured vertebra and reduces the incidence of severe complications caused by disability [2,3,4]. However, after PVP or PKP, recurrent VCFs at the index level have been reported, which are accompanied by progressive kyphosis. Moreover, several reports have noted the appearance of new fractures in adjacent segments, even without the occurrence of additional traumatic events [5,6]. Gu et al. [7,8] proposed the use of minimally invasive short-segment pedicle screw fixation combined with PVP or PKP to stabilize the fractured spine and prevent recurrent VCFs. However, older osteoporotic patients treated using pedicle screws experience higher rates of complications than do younger populations, including instrument loosening, pullout, and migration [9]. One method of resolving these issues is to use cortical bone trajectory (CBT) screws. CBT screws represent a novel spinal surgical technique that was first described by Santoni et al. [10] in 2009. Compared to the traditional pedicle screw trajectory, CBT screws have better pullout strength and may thereby solve the shortcomings associated with pedicle screws. Nevertheless, the usefulness of CBT screws in PVP for older patients with osteoporotic thoracolumbar VCFs remains unclear.

In this article, we assess the effectiveness of using short-segment CBT screw fixation with PVP in 12 patients with osteoporotic thoracolumbar VCFs in terms of patient outcomes, complications, and instrument failure. Notably, all patients achieved adequate pain relief and no patient experienced recurrent compression fractures or progressive kyphosis for at least 24 months following the procedure.

## 2. Case Series

Since October 2015, we began treating osteoporotic thoracolumbar junction VCFs patients using short-segment CBT instrumentation with vertebroplasty. We offered this treatment for all thoracolumbar junction osteoporotic compression fracture patients with good general condition and low anesthesiologic risk, without a trial of conservative treatment. No bone graft was used during surgery. Of the first 12 patients who were followed for >24 months, 2 were male and 10 were female. The average age was 74 years (range: 51 to 89 years), and the average T-score from dual-energy x-ray absorptiometry analysis was −3.6 (range: −2.5 to −4.8) (Table 1). The visual analog scale (VAS) pain scores significantly improved from 8 to 2.5 (paired t-test, *p* < 0.001) after surgery. The average blood loss was 36.3 mL (range: 25.0 to 50.0 mL). All participants regained normal ambulation post-surgery and none of the patients sustained any neurological deficits. The average sagittal Cobb’s angle significantly increased from 15.4° preoperatively to 18.8° (paired *t*-test, *p* = 0.007) at 24 months postoperatively. There was no radiographic or symptomatic evidence of recurrent fractures or instrument failure during the follow-up period. Besides, none of these patients received subsequent surgery to remove the implants to date.

## 3. Case illustration

A 53-year-old woman with a T-score of −4.8 (Figure 1A) according to dual-energy x-ray absorptiometry analysis, the worst of the series, suffered from severe back motion pain after slipping and falling. Her pain was partially relieved by analgesics and bed rest. Lumbar spine plain films revealed an L1 wedge deformity. Moreover, lumbar magnetic resonance imaging performed using the short tau inversion recovery sequence revealed L1 bone marrow edema indicative of an acute compression fracture (Figure 1B). After thoroughly explaining the planned surgical approach to the patient, we performed T12 and L2 CBT instrumentation with L1 vertebroplasty on this patient.

## 4. Technique Demonstration

During the surgery, the patient was placed in the prone position. We drew lines at the entry points at each level under C-arm localization, and an incision was made between the T12 and L2 lines. After making a skin and fascia incision, we dissected the muscles bilaterally. Under fluoroscopic guidance in the anterior-posterior view, a pilot hole was made using the sharp tip of a cannulated awl at the T12 left pars interarticularis; the awl was positioned at approximately five o’clock, and was aimed to the eleven o’clock point of the facet joint (Figure 2A). We subsequently adjusted the angulation to make the trajectory length approximately equal to the anterior portion of the awl and slowly tapped the tract until the junction of the slender part touched the pars. We then removed the cap with the wire and inserted the cannulated screw under the guiding pin (Figure 2B). The sagittal angle was adjusted in the cephalocaudal direction at L2 to minimize the length of the incision and the need for muscle dissection (Figure 2C). After four screws were inserted (Figure 2D,E), vertebroplasty with cement was performed under fluoroscopic guidance.

## 5. Postoperative Condition

Lumbar spine plain films were acquired the day after surgery (Figure 3A,B), and at 1, 6, 12, and 24 months postoperatively. After the surgery, the patient’s VAS score improved from 8 to 2, and she regained her ambulatory ability. At the 24-month follow-up evaluation, the patient showed no obvious changes in the kyphotic sagittal Cobb’s angle nor was there any indication of instrument loosening (Figure 3C–E).

## 6. Discussion

This case series assessed patient outcomes following short-segment CBT screw fixation with PVP, the presence of complications, such as screw loosening and re-fracturing, which are known to be associated with the more commonly used pedicle screws. For up to 24 months after the procedure, there were no instances of screw loosening, and the change in kyphotic sagittal Cobb’s angle was minimal, indicating that the contact strengths of the instruments remained intact. Collectively, our findings show that CBT screws were well tolerated in all patients, yielded good patient outcomes, and had no overt complications for at least 24 months after the procedure.

In their initial report, Santoni et al. [9] described the “cortical bone trajectory,” a novel lumbar pedicle screw placement approach that uses a mediolateral- and caudocephalad-directed path. Biomechanical analyses of the uniaxial pullout strength in cadaveric lumbar spines demonstrated that CBT screws exhibited a 30% greater pullout strength than did traditional pedicle screws [10]. In an in vivo analysis, the insertion torque, which is correlated with screw stability, was 1.7 times higher than traditional transpedicle screws [11]. Given that CBT screws exhibit a higher cortical bone density interface and their fixation is independent of the bone trabecular quality, they could serve as an alternative option for patients with osteoporosis.

Nevertheless, surgeons performing the CBT procedure described drawbacks, including the need to use intraoperative multi-planar fluoroscopy or a navigation system, which may not always be available. Consequently, to achieve an accurate trajectory in the absence of a navigation system, the surgeons were often required to change the fluoroscopy gantry angle, which in turn resulted in more radiation exposure. To address these issues, Huang et al. [12] reported a minimally invasive technique in midline lumbar interbody fusion and proposed the use of a percutaneous technique to insert CBT screws, thereby reducing the surgeon’s radiation exposure.

In our procedure, the cranial screws followed the typical lateral-cranial direction. However, the caudal screws were inserted in a slightly lateral-caudal direction in order to minimize the length of the wound and the need for muscular dissection. Mastukawa et al. [13] reported that the lateral-caudal, lateral-parallel, and lateral-cranial directions increase the screw pull-out strength by 6.1%, 21.1%, and 34.7%, respectively, compared with the traditional trajectory of transpedicle screws. Therefore, we believe inserting the screws in the lateral-caudal direction achieves better strength than the traditional trajectory for transpedicle screws. The trajectory of percutaneous pedicle screw fixation technique announced by Gu et al. [6,7] goes the traditional pedicle screw trajectory and carries the risks of instruments failure. Although CBT screw fixation needs a midline incision and multifidus muscle dissection, but the damage to multifidus muscle could be minimized with blunt dissection and only limited at the lamina of the fractured level. Besides, due to the minimal invasive approach and limited bony structure destruction with cement augmentation to the fractured vertebral bodies, the used of bone graft is not necessary.

There are additional advantages to this surgical technique. When encountering a new VCF at the instrumented segment, this technique allowed us to perform PVP to stabilize the vertebra easily because the trajectory differed from the CBT and trajectory for vertebroplasty. Moreover, if the vertebral fracture or the cement leakage led to cord compression, decompression could be performed directly with the midline incision.

The present study is limited by several factors, namely, retrospective nature, small sample size, lack of control group, and we did not directly compare CBT screws with traditional screws for validating the efficacy of these two techniques. And the procedure is a non-fusion instrumentation procedure. For thoracolumbar spinal trauma, some studies showed that this procedure wound be safe even in long term, but little is known about the non-fusion instrumentation for elderly patients and longer follow-up will be needed. Nevertheless, our results support that the CBT is safe and effective, and warrants further clinical study.

## 7. Conclusions

Owing to an aging global population, osteoporotic VCFs, a condition that any spine surgeon could encounter, represent an important health issue. We propose that the combination of CBT instrumentation with vertebroplasty is an efficacious and well-tolerated procedure for patients with osteoporotic thoracolumbar VCFs. We show that this surgical approach can alleviate the patients’ pain, minimize the change of kyphotic sagittal Cobb’s angle, maintain a low risk of instrument failure, and facilitate the ability to directly decompress the neural components. Moreover, no patient in our case series presented with a recurrent fracture. We therefore suggest that this technique is a promising alternative procedure that should be considered by spine surgeons when addressing osteoporotic thoracolumbar VCFs.

## Figures and Tables

**Figure 1 medicina-56-00082-f001:**
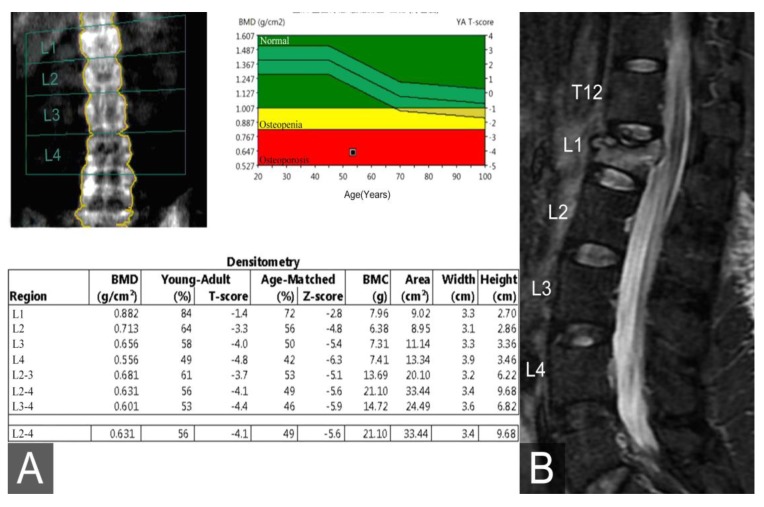
(**A**) The worst T-score on dual-energy x-ray absorptiometry analysis was −4.8 at L4. (**B**) The sagittal view of the lumbar spine magnetic resonance image acquired using the short tau inversion recovery sequence showed a hyper-intensity at L1, indicative of acute compression fracture-related bone marrow edema.

**Figure 2 medicina-56-00082-f002:**
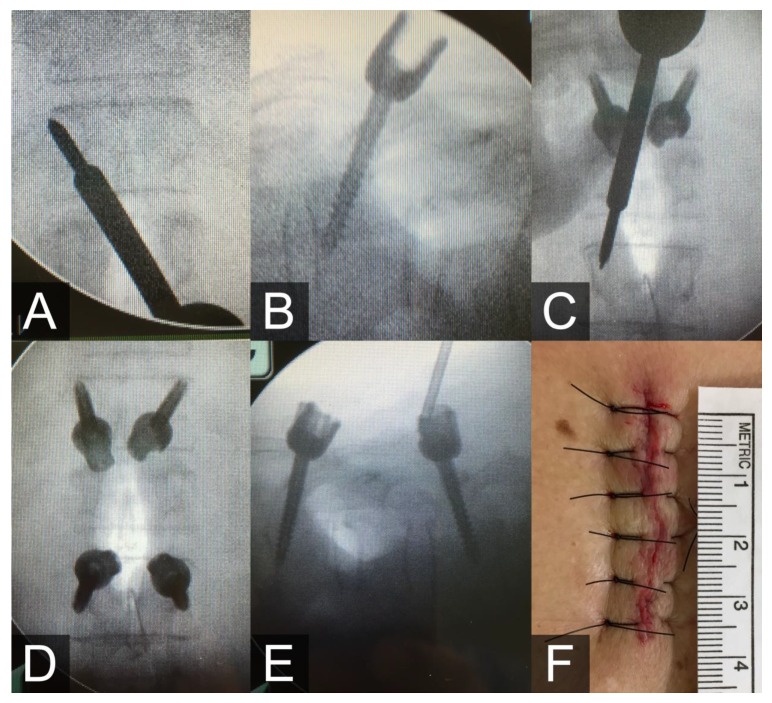
(**A**) The cannulated awl aimed from the five o’clock to the eleven o’clock point of the left pedicle of T12. (**B**) The inserted cannulated screws of T12. (**C**) The trajectory angulation was adjusted slightly in the lateral-caudal direction at L2. (**D**,**E**) The anterior-posterior view (**D**) and lateral view (**E**) after screws were inserted in the typical inferior to superior direction at T12 and in the superior to inferior direction at L2. (**F**) The measured wound length was 4 cm.

**Figure 3 medicina-56-00082-f003:**
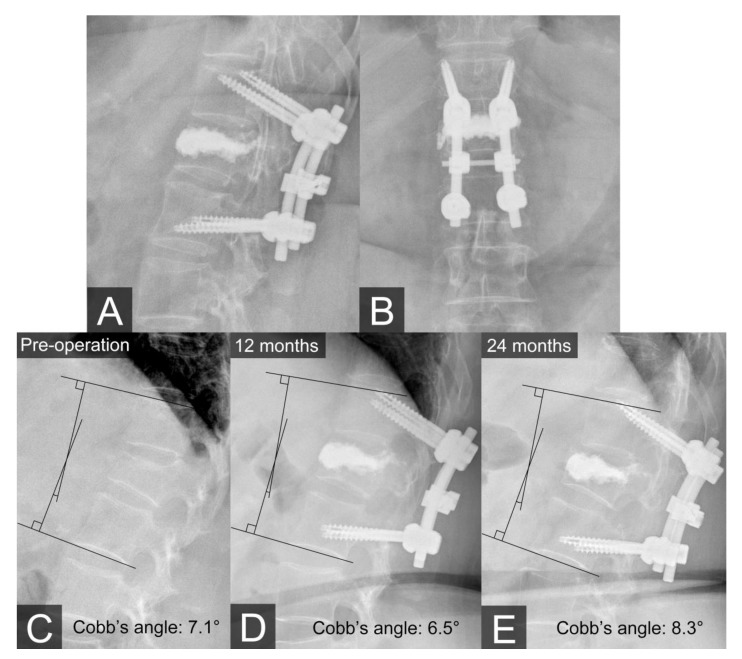
(**A**,**B**) Plain films on the first day after the operation. (**C**–**E**) Plain films showing the kyphotic sagittal Cobb’s angle preoperatively and at 12 months and 24 months postoperatively. There was no obvious change in the kyphotic sagittal Cobb’s angle or evidence that the screws loosened at the 24-month follow-up evaluation.

**Table 1 medicina-56-00082-t001:** Patient demographics of the cohort.

Case No.	Sex/Age	Level of Fracture	Blood Loss	T Score	VAS Change *	Sagittal Cobb’s Angle †
1	F/82	T12	35mL	−4.1	8 to 3	19° to 22°
2	M/65	L1	50mL	−3.2	7 to 2	12° to 12°
3	F/89	T11	40mL	−4.0	8 to 2	31° to 45°
4	F/78	T12	25mL	−3.4	9 to 2	11° to 13°
5	F/53	L1	30mL	−4.8	8 to 2	7° to 8°
6	M/51	L1	50mL	−2.5	8 to 3	16° to 18°
7	F/77	T12	50mL	−3.8	7 to 2	20° to 22°
8	F/81	L1	25mL	−3.1	9 to 3	9° to 13°
9	F/72	L1	25mL	−2.8	8 to 2	13° to 15°
10	F/84	L1	40mL	−4.5	8 to 3	15° to 19°
11	F/80	T12	30mL	−3.8	7 to 3	8° to 11°
12	F/78	L1	35mL	−3.6	9 to 3	24° to 28°

VAS, Visual analog scale. * VAS is comparison of preoperatively to 2 days postoperatively. † Change of sagittal Cobb’s angle from preoperatively to 24 months postoperatively.

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
