# Peer review of "Cortical Bone Trajectory Instrumentation with Vertebroplasty for Osteoporotic Thoracolumbar Compression Fracture"

_medicina, 2020, doi:10.3390/medicina56020082_

Round 1

Reviewer 1 Report

This is a case report analysis which assesses the effectiveness of using short-segment CBT screw fixation with PVP 66 in 12 patients with osteoporotic thoracolumbar VCFs. 

The first 12 cases using this new technique were reported. Descriptive statistics were used to report outcomes and complications. They further describe a specific case in detail and provide very detailed tables and picturesThis is a very novel approach to treatment and will serve as a stepping stone for future research. It is well written and very clear. The authors appropriately acknowledge the limitations to this case report. The main question is properly addressed.

Author Response

Thanks for your thorough review and kindly response. 

Reviewer 2 Report

Abstract:

The number of n=12 is quite low.

Introduction:

line 50: If you state that PVP or PKP "stabilizes" the vertebral body, do you have (biomechanical) proof for that? If so, please state.

line 54: There are also critical voices that need to be heard: https://doi.org/10.1002/jbmr.3651

line 65: Please add a reference that pedicle screw fixation reduces occurence of adjacent fractures.

Case series:

Is the range SD or SEM?

Why was Cobb's angle improved? Did you try to improve a coronar deformity by dorsal spondylodesis? How without removing the facet joints (e.g. via VCR)? If you mean the sagittal Cobb's angle, please name it like that.

Did you perform CT analysis to detect screw loosening?

What about osteoporosis treatment? In your cases, it seems absolutely fundamental.

Technique demonstration:

Please show another strictly lateral intraoperative x-ray of the screws and their angulation, especially with the cement.

Why did you not augment the screws?

Postoperative condition:

Initial screw loosening cannot be sufficiently detected in X-ray.

Why did you use a cross connector? Any data for that? Why did you not use longer screws since the length seems to create stability (https://www.ncbi.nlm.nih.gov/pubmed/31309769).

Discussion:

Did you assess VAS after longer than 2 days postoperatively? Where is the control group (especially due to the changes in VAS). One would be needed with "traditional" screw insertion and one without operative treatment.

Is the trajectory more important than the length or thickness of the screw?

How often did you detect cement leakage?

Conclusion:

Did you detect adjacent fractures by MRI? And the long-term VAS compared to conservative treatment?

Author Response

We greatly appreciate the efforts of the Editorial Board and reviewers for giving us the valuable comments on our manuscript. Included below is a point-by-point description of our responses to the reviewer’s comments.

1 The number of n=12 is quite low.

Response: We admit n=12 is quite low, it still have to establish a multi-center study to verify. Although, we provide a new insight into efficacy of combined CBT and VP for treating osteoporotic thoracolumbar fractures in the study.

Introduction:

1: line 50: If you state that PVP or PKP "stabilizes" the vertebral body, do you have (biomechanical) proof for that? If so, please state.

Response: As reviewer’s kindly suggestion, we have cited the reference in introduction section (Line 51, paragraph 1, page 2).

2.line 54: There are also critical voices that need to be heard: https://doi.org/10.1002/jbmr.3651

Response: We admit the outcomes between conservative treatment and surgical intervention is similar for the long-term. However, the rapid pain relief could be achieved by PVP or PKP.

3.line 65: Please add a reference that pedicle screw fixation reduces occurence of adjacent fractures.

Response: We didn’t mentioned about the reduced incidence of "adjacent" fracture by using screws fixation but the decreased recurrent fracture rate at the "index level". The reference has been cited as 7 and 8.

Case series:

1.Is the range SD or SEM?

Response: Standard deviation.

2.Why was Cobb's angle improved? Did you try to improve a coronar deformity by dorsal spondylodesis? How without removing the facet joints (e.g. via VCR)? If you mean the sagittal Cobb's angle, please name it like that

Response: The Cobb’s angle improved because the cement partial restore the fracture vertebral and the screws gave the effort to maintain the alignment. We didn’t try to improve a coronar deformity by dorsal spondylodesis, and the Cobb’s angle we mentioned was the sagittal Cobb’s angle. As reviewer's advise, we have corrected it in the manuscript.

3.Did you perform CT analysis to detect screw loosening?

Response: No.

4.What about osteoporosis treatment? In your cases, it seems absolutely fundamental.

Response: All of these patients underwent osteoporotic agent therapy after diagnosed, such as raloxifene, bisphosphate or teriparatide injection.

Technique demonstration:

1.Please show another strictly lateral intraoperative x-ray of the screws and their angulation, especially with the cement.

Response: This manuscript is emphasized on the minimal invasive instrumentation to augment the T-L junction fracture, therefore our pictures were focused on the small incision the direction of screws insertion.

Why did you not augment the screws?

Response: Since CBT screw was inserted into the cortical bone, it’s difficult to augment the screw via cement injection.

Postoperative condition:

1.Initial screw loosening cannot be sufficiently detected in X-ray.

Response: The reference of diagnosis approach regarding screw loosening:

DOI: 10.1302/0301-620x.86b3.14323

2.Why did you use a cross connector? Any data for that?

Response: In this patient, cement leakage was suspected on intra-operative fluorescence, therefore I performed laminectomy to check the dura sac and root, so I used a cross connector to maintain the stability.

3.Why did you not use longer screws since the length seems to create stability

Response: Cortical bone trajectory generally uses 30 to 40 mm screw length.

Discussion:

1.Did you assess VAS after longer than 2 days postoperatively? Where is the control group (especially due to the changes in VAS). One would be needed with "traditional" screw insertion and one without operative treatment.

Response: For rapid acute pain relief, we performed intervention in nearly every individual. Therefore, we can’t compare VAS of patients with conservative treatment.

2. One would be needed with "traditional" screw insertion and one without operative treatment.

Response: We generally do not perform traditionally screw insertion in these patients because of the destruction of the musculatures. And that’s why we chose CBT, a minimal invasive fixation technique.

3.Is the trajectory more important than the length or thickness of the screw?

Response: Yes, you got the point. The trajectory is more important than the length and thickness of the screw.

4.How often did you detect cement leakage?

Response: Not really often,  because I always injected cement under real-time fluorescence.

Conclusion:

1. Did you detect adjacent fractures by MRI? And the long-term VAS compared to conservative treatment?

Response: We didn't arrange MRI exam on patient since there’s neither clinical re-fracture symptoms such as acute severe back pain nor image evidence of implants failure.

2.And the long-term VAS compared to conservative treatment?

Response: We didn’t do that ever. However, the comparison study is a good idea for further investigation.   

Reviewer 3 Report

In this manuscript, the authors introduced a surgical technique that was a combination of cement vertebroplasty and posterior instrumentation cortical bone trajectory. Although this manuscript does not contain any comparisons or control group, the topic they investigated would be of interest for readers. I have some concerns as follows:

In introduction, the authors referred Gu et al.'s studies. In their studies, percutaneous pedicle screw (PPS) insertion was used. One advantage of PPS over open technique is less invasiveness to back muscle structure. It is known that preserving the origin of multifidus muscle at the spinous process is important, but the technique the authors used inevitably detach the multifidus muscle from the spinous process. Although this technique is less invasive than conventional posterior fusion, this is one of the disadvantages of this technique and should be discussed in the discussion section.   In method, please specify the indication for this procedure, since most VCFs have good results with conservative treatments and the indication for surgery should be limited to certain less prevalent conditions such as pseudarthrosis or neurological deficit. It is not the standard of care in most countries to operate on all acute VCFs. The authors did not mention about bone grafting. I believe secure long-term fusion can be obtained only with bony fusion. Instrumentation can provide only temporally stability until bony fusion and instrumentation without bony fusion cannot last long. I understand non-fusion instrumentation is often performed for young patients with certain types of spinal fracture, which requires implant removal after bony fusion, or patients with metastatic malignant lesions that the patient's life expectancy is limited. Do the patients undergo implant removal later? If so, wouldn't refracture at the removed site be a concern? If not, are there any concerns in patients in their 50s with this technique (2 cases)?  These questions should be answered in the discussion section.        

Author Response

We greatly appreciate the efforts of the reviewer for giving us the valuable comments on our manuscript. Included below is a point-by-point description of our responses to the reviewer’s comments.

Comment 1: Although this technique is less invasive than conventional posterior fusion, this is one of the disadvantages of this technique and should be discussed in the discussion section.

Response: As reviewer's recommendation, we have mentioned in the discussion section(Line 163-167, Paragraph 4, Page 6).

Comment 2: In method, please specify the indication for this procedure, since most VCFs have good results with conservative treatments and the indication for surgery should be limited to certain less prevalent conditions such as pseudarthrosis or neurological deficit.

Response: As reviewer's kindly advise, We have advocated the indication as osteoporotic "thoracolumbar junction" compression fracture in the case series(Line 71, Paragraph 1, Page 2).

Comment 3: It is not the standard of care in most countries to operate on all acute VCFs. The authors did not mention about bone grafting.

Response: As reviewer's suggestion, we have mentioned in the Case series(Line 2, Paragraph 1, Page 2) and discussion section(Line 167-169, Paragraph 4, Page 6)

Comment 4:I believe secure long-term fusion can be obtained only with bony fusion. Instrumentation can provide only temporally stability until bony fusion and instrumentation without bony fusion cannot last long. I understand non-fusion instrumentation is often performed for young patients with certain types of spinal fracture, which requires implant removal after bony fusion, or patients with metastatic malignant lesions that the patient's life expectancy is limited. Do the patients undergo implant removal later? 

Response: None of screw was removed even in young patient, and we have described in case series section (Line 81-82, Paragraph 2, Page 2).

Round 2

Reviewer 2 Report

We thank the authors for the improvements of the manuscript.

3.Is the trajectory more important than the length or thickness of the screw?

Response: Yes, you got the point. The trajectory is more important than the length and thickness of the screw.

-> Please give a (biomechanical) reference of that.

4.How often did you detect cement leakage?

Response: Not really often,  because I always injected cement under real-time fluorescence.

-> And in the postoperative x-ray? In literature, up to 11-73% of cement leakage are reported. In fluoroscopy, just 34% in lateral X-ray and 48% inlateral and AP view of leakage can be assessed. Therefore, it is ignorant to just use fluorescence.

The lack of a control group is a major limitation and needs to be stated.
It is important to assess VAS or Oswestry Disability Index after more than two years.

Author Response

3.Is the trajectory more important than the length or thickness of the screw?

Response: Yes, you got the point. The trajectory is more important than the length and thickness of the screw.

-> Please give a (biomechanical) reference of that.

Response:

We followed professor Matsukawa's suggestion and used screws 5.5mm in diameter and 40 mm in length for CBT. The trajectory itself mainly located in the cortical bone of the vertebrae and brought out better fixation strength than TPS trajectory, even the TPS screws were longer and thicker than CBT screws.

Information: https://doi.org/10.1097/BSD.0000000000000258

4.How often did you detect cement leakage?

Response: Not really often,  because I always injected cement under real-time fluorescence.

-> And in the postoperative x-ray? In literature, up to 11-73% of cement leakage are reported. In fluoroscopy, just 34% in lateral X-ray and 48% inlateral and AP view of leakage can be assessed. Therefore, it is ignorant to just use fluorescence.

Response:

We admit your opinion and our approach could provide immediate decompression to check the neural elements while cement leakage was suspected during operation.

The lack of a control group is a major limitation and needs to be stated.
It is important to assess VAS or Oswestry Disability Index after more than two years.

Response: As reviewer's kindly recommendation, we add the limitation in discussion section (Line 183-184, Paragraph 6, Page 6). In the present study, patients are stable for the long-term (24 months). Supposedly as a result of small sample size as you said before, and we will take the valuable suggestion for the further study. Honestly, we already perform comparison study between CBT and TPS. Hope to share with you if you received our work.

Reviewer 3 Report

The authors did not address my comments enough.

"It is not the standard of care in most countries to operate on all acute VCFs".

Are you operating on all acute osteoporotic "thoracolumbar junction" compression fracture? if not, how did you choose the surgical candidate ?

Please answer this question with reasons. 

"I understand non-fusion instrumentation is often performed for young patients with certain types of spinal fracture, which requires implant removal after bony fusion, or patients with metastatic malignant lesions that the patient's life expectancy is limited.  "

I did not talk about fusion at the fracture site, fusion between vertebrae by posterior or postro-lateral fusion. Usually posterior instrumentation is performed to ultimately obtain bony intervertebral fusion, except some conditions I mentioned. It is known the anterior part is still somewhat mobile even after posterior instrumentation and, especially for elderly population, a long term instrumentation only fixation without bony intervertebral fusion will lead to implant failure, fracture, or adjacent segmental diseases. For me, this method the author described seems highly likely to lead to these complications in longer term, regardless of fracture site condition. This procedure is actually a two level fusion skipping mid-vertebrae without bone graft, I do not believe no implant failure will be observed, and no spine surgeon in my country would think it is safe. 

Please provide any rationales with proper references indicating that this instrumentation only fixation for elderly population is safe for long-term. 

Author Response

"It is not the standard of care in most countries to operate on all acute VCFs".

Are you operating on all acute osteoporotic "thoracolumbar junction" compression fracture? if not, how did you choose the surgical candidate ?

Please answer this question with reasons. 

Response: 

Sorry, we didn't catch your point last time.

In patients with osteoporotic thoracolumbar junction compression fracture with good general condition and low risk of general anesthesia, we would suggest patients underwent CBT fixation with PVP. But if the patients carried high risk of intubation and general anesthesia, we wound perform PVP only. 

"I understand non-fusion instrumentation is often performed for young patients with certain types of spinal fracture, which requires implant removal after bony fusion, or patients with metastatic malignant lesions that the patient's life expectancy is limited.  "

I did not talk about fusion at the fracture site, fusion between vertebrae by posterior or postro-lateral fusion. Usually posterior instrumentation is performed to ultimately obtain bony intervertebral fusion, except some conditions I mentioned. It is known the anterior part is still somewhat mobile even after posterior instrumentation and, especially for elderly population, a long term instrumentation only fixation without bony intervertebral fusion will lead to implant failure, fracture, or adjacent segmental diseases. For me, this method the author described seems highly likely to lead to these complications in longer term, regardless of fracture site condition. This procedure is actually a two level fusion skipping mid-vertebrae without bone graft, I do not believe no implant failure will be observed, and no spine surgeon in my country would think it is safe. 

Please provide any rationales with proper references indicating that this instrumentation only fixation for elderly population is safe for long-term. 

Response:

In 2009, Dai LY, et al. has proposed an article regarding bone graft fusion was not necessary in patients with thoracolumbar burst fracture in the long-term follow-up study. The article information: 10.2106/JBJS.H.00510

Round 3

Reviewer 2 Report

The manuscript has been improved.

All concerns have been sufficiently addressed.

Author Response

Dear professor:

Thanks for your recommendation and kindly response. We will keep working hard on our further research.

Sincerely,

Chen Chao-Hsuan

Reviewer 3 Report

"In patients with osteoporotic thoracolumbar junction compression fracture with good general condition and low risk of general anesthesia, we would suggest patients underwent CBT fixation with PVP. But if the patients carried high risk of intubation and general anesthesia, we wound perform PVP only."

-> Describe this in the Case Series section like following:

"We offered this treatment for all acute TL-junction VCF patients with good general conditions and low anesthesiologic risk, without a trial of conservative treatment. We screened XX patients and XX met the criteria, and YY agreed to undergo this treatment."

"In 2009, Dai LY, et al. has proposed an article regarding bone graft fusion was not necessary in patients with thoracolumbar burst fracture in the long-term follow-up study. The article information: 10.2106/JBJS.H.00510"

-> The average patients' age of the abovementioned study was much younger than yours. The study results are not comparable.  I don't care about the non-fusion instrumentation for the younger population, but your population was completely different.

If you can automatically apply the principles obtained from young patients to elderly patients, we never need to think about specialized techniques, including this CBT+VP technique. Add a sentence regarding the potential risk for later failure due to the non-fusion procedure in the limitation section like the following:

"This procedure is a non-fusion instrumentation procedure. For TL spinal trauma, some studies showed that this procedure would be safe even in long term, but little is known about the non-fusion instrumentation for elderly patients Thus, longer follow-up will be needed...

Author Response

Dear professor:

Thanks for you recommendation and kindly response. We made modification of our manuscript as you recommended and wish it could reach your high standard and be accepted. The point-to-point response to the reviewer are listed below:

-> Describe this in the Case Series section like following:

1."We offered this treatment for all acute TL-junction VCF patients with good general conditions and low anesthesiologic risk, without a trial of conservative treatment. We screened XX patients and XX met the criteria, and YY agreed to undergo this treatment."

Response: As reviewer's suggestion, we add the description of candidates for the treatment.(Line 73-75, Paragraph 1, Page 2). 

2."In 2009, Dai LY, et al. has proposed an article regarding bone graft fusion was not necessary in patients with thoracolumbar burst fracture in the long-term follow-up study. The article information: 10.2106/JBJS.H.00510"

-> The average patients' age of the abovementioned study was much younger than yours. The study results are not comparable.  I don't care about the non-fusion instrumentation for the younger population, but your population was completely different.

If you can automatically apply the principles obtained from young patients to elderly patients, we never need to think about specialized techniques, including this CBT+VP technique. Add a sentence regarding the potential risk for later failure due to the non-fusion procedure in the limitation section like the following:

"This procedure is a non-fusion instrumentation procedure. For TL spinal trauma, some studies showed that this procedure would be safe even in long term, but little is known about the non-fusion instrumentation for elderly patients Thus, longer follow-up will be needed...

Response: As reviewer's suggestion, we add the citation to the discussion section (Line187-190, Paragraph 6, Page 6). 

Sincerely,

Chen Chao-Hsuan
